# Deep-Reinforcement-Learning-Based Joint Energy Replenishment and Data Collection Scheme for WRSN

**DOI:** 10.3390/s24082386

**Published:** 2024-04-09

**Authors:** Jishan Li, Zhichao Deng, Yong Feng, Nianbo Liu

**Affiliations:** 1Yunnan Key Laboratory of Computer Technology Applications, Kunming University of Science and Technology, Kunming 650500, China; 20212204267@stu.kust.edu.cn (J.L.); dengzhichao@stu.kust.edu.cn (Z.D.); 2School of Computer Science and Engineering, University of Electronic Science and Technology of China, Chengdu 611731, China; liunb@uestc.edu.cn

**Keywords:** wireless rechargeable sensor networks, unmanned aerial vehicles, deep reinforcement learning, route protocol

## Abstract

With the emergence of wireless rechargeable sensor networks (WRSNs), the possibility of wirelessly recharging nodes using mobile charging vehicles (MCVs) has become a reality. However, existing approaches overlook the effective integration of node energy replenishment and mobile data collection processes. In this paper, we propose a joint energy replenishment and data collection scheme (D-JERDG) for WRSNs based on deep reinforcement learning. By capitalizing on the high mobility of unmanned aerial vehicles (UAVs), D-JERDG enables continuous visits to the cluster head nodes in each cluster, facilitating data collection and range-based charging. First, D-JERDG utilizes the K-means algorithm to partition the network into multiple clusters, and a cluster head selection algorithm is proposed based on an improved dynamic routing protocol, which elects cluster head nodes based on the remaining energy and geographical location of the cluster member nodes. Afterward, the simulated annealing (SA) algorithm determines the shortest flight path. Subsequently, the DRL model multiobjective deep deterministic policy gradient (MODDPG) is employed to control and optimize the UAV instantaneous heading and speed, effectively planning UAV hover points. By redesigning the reward function, joint optimization of multiple objectives such as node death rate, UAV throughput, and average flight energy consumption is achieved. Extensive simulation results show that the proposed D-JERDG achieves joint optimization of multiple objectives and exhibits significant advantages over the baseline in terms of throughput, time utilization, and charging cost, among other indicators.

## 1. Introduction

Benefiting from the advancements in wireless power transfer (WPT) technology [1,2], wireless rechargeable sensor networks (WRSNs) equipped with wireless charging systems have emerged as an effective solution for addressing node energy constraints [3,4]. Unlike traditional approaches [5,6,7], WRSN fundamentally overcomes the predicament of nodes relying solely on battery power. It provides a promising solution for the sustainable energy replenishment of nodes [8,9,10]. WRSN normally includes one or more mobile charging vehicles (MCVs) and a base station (BS) for the battery replacement of MCVs. MCV can move autonomously and is equipped with a wireless charging device to replenish energy for nodes by wireless charging. However, in remote and harsh environments, e.g., farmlands, forests, or disaster areas, it becomes challenging for ground vehicles to enter designated areas and perform close-range wireless charging [11,12,13]. Utilizing UAVs as aerial mobile vehicles to provide remote services to ground devices is considered to bring significant benefits to WRSN [14]. UAV possesses excellent maneuverability, coverage capabilities, and relatively low operating costs. It can perform various tasks within the network coverage, such as routing, communication, data collection, and energy supply [15,16,17,18]. In UAV-assisted wireless rechargeable sensor networks (UAV-WRSNs), UAVs can serve both as energy transmitters and data transceivers. By continuously accessing and exchanging data with nodes within their communication range, they can forward the data to the BS. This eliminates the energy consumption caused by multihop transmissions between nodes. Additionally, UAVs provide range-based charging to ensure simultaneous data collection and energy replenishment.

Although UAV offers new possibilities for the future development of WRSN, more research is needed to consider the unified process of node energy replenishment and data collection in UAV-WRSN. There are still unresolved issues in UAV-WRSN. For example, in a one-on-one node service scenario, UAV must hover multiple times, increasing hover time and energy consumption, leading to subsequent node data overflow or energy depletion. Moreover, target node selection is crucial to enable UAV to cover more nodes with each hover. In practical environments, there is a significant difference in energy consumption among nodes, and some nodes may run out of energy earlier due to heavy loads. Therefore, balancing node energy consumption and solving the problem of energy imbalance among nodes are equally important. Based on the above issues, this paper proposes D-JERDG, which integrates node joint energy replenishment and data collection using deep reinforcement learning.

The main contributions of this study can be summarized as follows:First, we consider deploying UAV in delay-tolerant WRSN and combining wireless power transfer (WPT) with wireless information transfer (WIT) technologies. To achieve the unified process of sensor node energy replenishment and mobile data collection, we propose a deep-reinforcement-learning-based method called D-JERDG for UAV-WRSN.We introduce a cluster head selection algorithm based on an improved dynamic routing protocol to minimize the number of UAVs hovering and balance node energy consumption. Based on the obtained cluster head visiting sequence, we employ the simulated annealing algorithm [19] to approximate the optimal solution for the traveling salesman problem (TSP), thereby reducing the UAV flight distance. We then employ the DRL model MODDPG [20] and design a multiobjective optimized reward function to control the UAV instantaneous speed and heading, aiming to minimize the node death rate and the UAV’s average energy consumption.Simulation results are conducted to evaluate the feasibility and effectiveness of D-JERDG. The results demonstrate that D-JERDG outperforms existing algorithms in node death rate, time utilization, and charging cost.

The remaining parts of this paper are as follows: Section 2 introduces related work, Section 3 describes the system model and problem formulation of UAV-WRSN, Section 4 presents the cluster head selection algorithm based on an improved dynamic routing protocol and the MODDPG algorithm in D-JERDG, Section 5 validates the effectiveness and feasibility of D-JERDG through comparative experiments, and Section 6 summarizes the paper and discusses future research directions.

## 2. Related Work

In this section, we provide a brief overview of the existing work in three relevant domains: cluster-based networks [21,22,23,24,25,26,27,28], traditional algorithm-based UAV trajectory planning [29,30,31,32,33,34,35,36,37], and DRL-based UAV trajectory planning [38,39,40,41,42,43].

In traditional WRSN, the one-to-one node charging mode can often lead to inefficient movement of MCV, resulting in wastage of energy. Moreover, the convergence nodes bear the primary data transmission tasks, leading to faster energy depletion and premature failure of nodes, thus affecting the overall network lifetime. To balance the energy consumption of nodes, the charging method often involves dividing the nodes into several clusters using a hierarchical clustering approach. Within each cluster, a few nodes closest to the base station (BS) are selected as cluster heads, and communication links are established between the cluster heads and the BS, such as LEACH [21], K-means [22], Hausdorff [23], and HEED [24]. These methods typically use a rotating cluster head approach to reduce energy consumption and extend network lifetime. For example, in [25,26,27,28], the authors studied the division of the network into multiple clusters using clustering algorithms. They employed a mobile charger (MC) to periodically replenish the energy of anchor nodes in each cluster according to the generated shortest visiting path. Wu et al. designed a joint solution that integrates both aspects and proposed a heuristic-based MC scheduling scheme to maximize the charging efficiency of the MC while minimizing its energy consumption [25]. Li et al. proposed an energy-efficiency-oriented heterogeneous paradigm (EEHP), which is a routing protocol based on multihop data transmission. It reduces the energy consumption for data transmission by employing multihop data transfer and shortens the charging distance for the MC [26]. Han et al. used the K-means clustering algorithm to divide the network into multiple clusters and proposed a semi-Markov model to update anchor nodes. They deployed two MCs that periodically moved in opposite directions to visit anchor nodes, charging the sensor nodes (SNs) within the charging range and collecting data from the cluster heads [27]. Zhao et al. focused on achieving joint optimization of efficient charging and data collection in randomly deployed WRSN. They periodically selected anchor points and arranged MCV to visit these locations and charge the nodes sequentially [28].

For sustainable monitoring, WRSN can be applied in remote and resource-limited areas, such as rural farmlands, forests, or disaster zones [29]. In such harsh environments, UAVs can be used as auxiliary aerial base stations to efficiently collect data and replenish node energy, which significantly benefits WRSN [30]. For example, in [31,32,33,34], researchers studied scheduling and designing UAV trajectories to improve the system charging efficiency. Xu et al. studied a new UAV-enabled wireless power transfer system to maximize the total energy received by considering the optimization of the trajectory of UAV under the maximum velocity constraint [31]. Liu et al. proposed UAV-WRSN and solved the subproblems of UAV scheduling and trajectory optimization separately. They aimed to minimize the number of UAV hover points, SN with repeated coverage, and UAV flight distance [32]. Wu et al. investigated the trajectory optimization problem for UAV in UAV-WRSN. They decomposed the problem of maximizing energy efficiency into an integer programming problem and a nonconvex optimization problem, effectively reducing the UAV flight distance and algorithm complexity and maximizing the energy utilization efficiency of the UAV [33]. Zhao et al. proposed an improved ant colony algorithm to plan UAV flight trajectories, achieving shorter flight paths and network lifetimes [34]. In [35,36,37], researchers explored using UAVs as data collectors and mobile chargers, simultaneously providing data collection and energy transfer to the nodes. Baek et al. considered node energy consumption and replenishment to maximize the WRSN lifetime. They optimized the UAV hover locations and durations to maximize the remaining energy of the nodes [35]. Lin et al. studied the collaboration between rechargeable sensors and UAVs to accomplish regular coverage tasks. They introduced a new concept of coverage called periodic area coverage, aiming to maximize the energy efficiency of the UAV [36]. Hu et al. formulated a nonconvex optimization problem to minimize the average age of information (AoI). They divided it into a time allocation problem and UAV trajectory optimization problem and solved it optimally using dynamic programming and ant colony algorithms [37].

DRL has proven to be an effective solution for decision-making problems on sequential data. It has been widely applied in various fields and has achieved notable results. In the context of UAV-WRSN, several studies are worth mentioning: Bouhamed et al. employed two reinforcement learning (RL) methods, namely, deep deterministic policy gradient (DDPG) and Q-learning (QL), to train UAV for data collection tasks. DDPG was utilized to optimize UAV flight trajectories in environments with obstacle constraints, while QL was used to determine the order of visiting nodes [38]. Liu et al. proposed a UAV path planning based on reinforcement learning, which enables the UAV to respond to the position change of the cluster nodes, reduces the flight distance and the energy consumption, and increases the time utilization ratio [39]. Li et al. proposed a flight resource allocation framework based on the DDPG algorithm. They optimized the UAV instantaneous heading, speed, and selection of target nodes. They utilized a state representation layer based on long short-term memory (LSTM) to predict network dynamics and minimize data packet loss [40]. Shan et al. presented a DRL-based trajectory planning scheme for multi-UAVs in WRSN. They established a network model for multi-UAV path planning. They optimized the network model using an improved hybrid energy-efficient distributed (HEED) clustering algorithm to obtain the optimal charging path [41]. Liu et al. considered WRSN assisted by UAVs and vehicles. In their study, UAVs served as mobile chargers to replenish energy for nodes, while mobile vehicles acted as mobile base stations to replace UAV batteries. The authors utilized a multiobjective deep Q-network (DQN) algorithm to minimize sensor downtime and optimize UAV energy consumption [42]. Wang et al. proposed a dynamic spatiotemporal charging scheduling scheme based on deep reinforcement learning, given the discrete charging sequence planning and continuous charging duration adjustment in mobile charging scheduling, to improve the charging performance while avoiding the power death of nodes [43]. Table 1 shows how the related work differs from our scheme.

## 3. System Model and Problem Formulation

In this section, we present the system model and problem formulation of UAV-assisted wireless rechargeable sensor networks (UAV-WRSNs). For the sake of clarity, Table 2 lists the symbols used in this paper.

### 3.1. System Model

#### 3.1.1. Network Model

As shown in Figure 1, the network is deployed in a two-dimensional area and consists of a UAV, a base station (BS), and randomly distributed sensor nodes. The UAV is assumed to have computing and wireless transmission capabilities, be equipped with a sufficiently charged battery, and be able to obtain its position through the Global Positioning System (GPS). The BS has ample energy and communication capabilities, enabling direct wireless communication with the UAV. The sensor nodes can monitor their remaining energy and data fusion capabilities. Cluster member (CM) nodes send data packets to cluster head (CH) nodes, which aggregate the data and store them in a data buffer. The UAV takes off from the BS and follows a “fly-hover” pattern to visit each CH node in a predetermined order continuously. The CH nodes consume energy to upload data packets to the UAV, while the UAV provides charging to the nodes within its coverage range using radio frequency transmission technology.

#### 3.1.2. Sensor Node Model

The network deploys *k* sensor nodes with fixed and known positions. Consider the set of nodes in the network as $S_{x}≜\left\{s_{1},s_{2},s_{3},\ldots ,s_{k}\right\}$. The CH nodes are denoted as $ch_{i}≜\left\{ch_{1},ch_{2},ch_{3},\ldots ,ch_{m}\right\}$, and the CM nodes are denoted as $cm_{j}≜\left\{cm_{1},cm_{2},cm_{3},\ldots ,cm_{h}\right\}$. Assuming that node batteries can be charged within a very short time compared with the data collection time by the UAV, we can neglect the charging time. Regarding the energy consumption of nodes in sleep mode, since this portion of energy consumption is very small and can be considered negligible, we have not included it as a significant influencing factor, and therefore, it has not been incorporated into the node energy consumption. Nodes only consume energy when receiving and transmitting data. The data buffer length of the CH node $ch_{i}$ at *t* is denoted as $D_{ch_{i}}\left(t\right)=\left[0,Q_{max}\right]$. Based on the literature [43], assuming that, at any time *t*, the energy consumption of the CM node $cm_{j}$ sending a unit data packet (where the unit data packet size *f* is 1024 kb) to the CH node $ch_{i}$ can be given as follows:(1)$$E_{j}^{cm}\left(t\right)=e_{t}\sum_{j=1}^{h}f_{j,i}$$
where $e_{t}$ indicates the energy consumption of transmitting unit data, the energy consumption of the CH node $ch_{i}$ can be divided into two parts: reception energy and transmission energy:(2)$$E_{i}^{ch}\left(t\right)=\sum_{j=1}^{h}f_{j,i}\sum_{i=1}^{m}e_{r}+E_{ch_{i}}^{t}\left(t\right)$$
where $e_{r}$ indicates the energy consumption of receiving unit data, and the remaining energy of the CM node $cm_{j}$ and the CH node $ch_{i}$ at *t* can be respectively defined as follows:(3)$$RE_{j}^{cm}\left(t\right)=E_{j}^{cm}\left(t-1\right)-E_{j}^{cm}\left(t\right)$$
(4)$$RE_{i}^{ch}\left(t\right)=E_{i}^{ch}\left(t-1\right)-E_{i}^{ch}\left(t\right)$$

#### 3.1.3. UAV Model

The UAV maintains a fixed flight altitude of *H*. It acquires the positions of all nodes before taking off from the base station and maintains communication with the base station throughout the operation. At *t*, the three-dimensional coordinates of the UAV can be represented as $\left[x_{u}\left(t\right),y_{u}\left(t\right),H\right]$. The distance between the UAV and the node plays a crucial role in determining the link quality and the ability of the node to upload data. In our scenario, we assume that the UAV has limited communication and charging ranges, denoted as $R_{d}$ (maximum data transmission radius) and $R_{c}$ (maximum charging radius), respectively. $\Delta d_{u,ch_{i}^{tar}}$ represents the distance between the UAV and the target CH node if $\Delta d_{u,ch_{i}^{tar}}\leq R_{d}$; the UAV enters the hovering state to collect data from the target CH node and charge all nodes within the charging range. The instantaneous speed $v_{u}\left(t\right)$ and the instantaneous heading angle $\theta_{u}\left(t\right)$ describe the flight control of the UAV. For safety reasons, $v_{u}\left(t\right)$ must be within the minimum and maximum speed limits.
(5)$$V_{\min}\leq v_{u}\left(t\right)\leq V_{\max}$$

This paper employs a rotor-wing UAV, and the energy consumption of the UAV can be divided into propulsion energy consumption and communication energy consumption. The propulsion energy consumption is further divided into flight energy consumption and hovering energy consumption [20]. When the UAV has an instantaneous speed $v_{u}\left(t\right)$, the propulsion energy consumption model of the UAV can be given as follows:(6)Eupro(vu(t))=P0(1+3vu(t)2Utip2)+Pf(1+vu(t)44v04−vu(t)22v02)12+12d0ρairsrotorArotorvu(t)3
where $P_{0}$ and $P_{f}$ are constants that represent the blade profile power and induced power in hovering status, respectively; $U_{tip}$ indicates the tip speed of the rotor blade; $v_{0}$ denotes the mean rotor-induced speed; and $d_{0}$, $\rho_{air}$, $s_{rotor}$, and $A_{rotor}$, respectively, denote the fuselage drag ratio, air density, rotor solidity, and rotor disc area. At t, UAV speed is $v_{u}\left(t\right)$, and the flight energy consumption is expressed as $E_{u}^{fly}=E_{u}^{pro}\left(v_{u}\left(t\right)\right)$. It is worth noting that the UAV is in hovering status with flight speed $v_{u}\left(t\right)=0$ and hovering energy consumption $E_{hover}=E_{u}^{pro}\left(v_{u}\left(t\right)=0\right)$. To facilitate the expression, we divide the time domain *T* into *n* time steps, where each time step is denoted as $t_{0},t_{1},t_{2},\ldots t_{n}$; thus, the total flight energy consumption of the UAV at each time step $t_{n}$ is given as follows:(7)$$E_{u}^{total}\left(t_{n}\right)=\int_{0}^{t_{n}}E_{u}^{pro}\left(v_{u}\left(t\right)\right)dt$$
the UAV’s average flight energy consumption is given as follows:(8)$$E_{u}^{ave}=\frac{E_{u}^{total}\left(T\right)}{T}$$

#### 3.1.4. Transmission Model

The wireless communication link between the UAV and the nodes mainly consists of a line-of-sight (LoS) link and a non-line-of-sight (NLoS) link, similar to many existing works [20,39,40]. Let the coordinate of the target CH node $ch_{i}^{tar}$ be $\left[x_{ch_{i}^{tar}},y_{ch_{i}^{tar}},0\right]$; the free path loss of the LoS link between the UAV and $ch_{i}^{tar}$ at *t* is modeled as follows:(9)$$\begin{array}{cc} h_{u,ch_{i}^{tar}}\left(t\right) & =\gamma_{0}\Delta d_{u,ch_{i}^{tar}}{\left(t\right)}^{-2}\\ \; & =\frac{\gamma_{0}}{H^{2}+{\left(x_{u}\left(t\right)-x_{ch_{i}^{tar}}\right)}^{2}+{\left(y_{u}\left(t\right)-y_{ch_{i}^{tar}}\right)}^{2}} \end{array}$$
where the channel’s power gain at a reference distance of $\Delta d_{u,ch_{i}^{tar}}=1$ m is denoted by $\gamma_{0}$, and $\Delta d_{u,ch_{i}^{tar}}\left(t\right)$ denotes the Euclidean distance between the UAV and $ch_{i}^{tar}$ at *t*, where the path loss of the NLoS link is given as $\eta^{NLos}\gamma_{0}\Delta d_{u,ch_{i}^{tar}}{\left(t\right)}^{-2}$, and $\eta^{NLos}$ is the attenuation coefficient of the NLoS links.

At *t*, the LoS link probability between the UAV and $ch_{i}^{tar}$ is modeled as follows:(10)$$P_{u,ch_{i}^{tar}}^{LoS}\left(\theta_{ch_{i}^{tar}}\left(t\right)\right)=\frac{1}{1+\alpha \exp(-\beta \left(\theta_{u,ch_{i}^{tar}}\left(t\right)-\alpha \right))}$$
where α and β are constant sigmoid parameters that depend on the signal propagation environment and propagation distance. $\theta_{ch_{i}^{tar}}\left(t\right)$ is the elevation angle of the UAV and $ch_{i}^{tar}$ in degree; it is given as $\theta_{ch_{i}^{tar}}\left(t\right)=\frac{180}{\pi }\sin^{-1}\left(\frac{H}{\Delta d_{u,ch_{i}^{tar}}}\right)$. The NLoS link probability is given as follows:(11)$$P_{u,ch_{i}^{tar}}^{NLoS}\left(\theta_{ch_{i}^{tar}}\left(t\right)\right)=1-P_{ch_{i}^{tar}}^{LoS}\left(\theta_{ch_{i}^{tar}}\left(t\right)\right)$$
let $g_{u,ch_{i}^{tar}}\left(t\right)$ be the wireless channel gain between the UAV and $ch_{i}^{tar}$; it is modeled as follows:(12)$$\begin{array}{cc} g_{u,ch_{i}^{tar}}\left(t\right) & =\left(P_{u,ch_{i}^{tar}}^{LoS}\left(\theta_{u,ch_{i}^{tar}}\left(t\right)\right)+\eta^{NLos}P_{u,ch_{i}^{tar}}^{NLoS}\left(\theta_{u,ch_{i}^{tar}}\left(t\right)\right)\right)h_{u,ch_{i}^{tar}}\left(t\right) \end{array}$$
in this paper, we assume that the uplink and downlink channels are approximately equal. As a result, the channel power gain between the UAV and $ch_{i}^{tar}$ is given as follows:(13)$$\begin{array}{cc} h_{u,ch_{i}^{tar}}\left(t\right)\approx g_{u,ch_{i}^{tar}}\left(t\right) & =(P_{u,ch_{i}^{tar}}^{LoS}\left(\theta_{u,ch_{i}^{tar}}\left(t\right)\right)\\ \; & +\eta^{NLos}P_{u,ch_{i}^{tar}}^{NLoS}\left(\theta_{u,ch_{i}^{tar}}\left(t\right)\right))h_{u,ch_{i}^{tar}}\left(t\right) \end{array}$$

Assuming that the UAV establishes a communication link with $ch_{i}^{tar}$ at *t* and $ch_{i}^{tar}$ maintains a constant transmit power $P_{t}$ to upload data to the UAV, according to Shannon’s formula [44], the data transmission rate between the UAV and $ch_{i}^{tar}$ can be given as follows:(14)$$R_{ch_{i}^{tar}}^{t}\left(t\right)=B\log_{2}\left(1+\frac{P_{t}{\left|g_{u,ch_{i}^{tar}}\left(t\right)\right|}^{2}}{\sigma^{2}}\right)$$
where *B* and $\sigma^{2}$ represent the channel bandwidth and noise power, respectively, and $P_{t}$ denotes the transmit power of the node; hovering time (upload data time) is given as follows:(15)$$t_{hover}=\frac{D_{ch_{i}^{tar}}\left(t\right)}{R_{ch_{i}^{tar}}^{t}\left(t\right)}$$
the energy consumption for $ch_{i}^{tar}$ to upload data at time *t* can be given as follows:(16)$$E_{ch_{i}^{tar}}^{t}\left(t\right)=P_{t}t_{hover}$$

### 3.2. Problem Formulation

In the proposed D-JERDG, we consider the UAV capabilities of energy replenishment and data collection. The UAV departs from the BS and moves according to a planned flight path. It is responsible for collecting data generated by all cluster heads in the network. After completing a mission cycle, the UAV returns to the BS to recharge and transmit the collected data back to the BS. The flight path consists of the hovering positions and the CH node access sequence. First, to improve data collection efficiency, the UAV needs to serve as many nodes as possible within the maximum flight time *T* and collect data. Second, the charging strategy for the target CH node is crucial to minimize the node death rate and flight costs. To ensure that high-energy-consuming nodes and remote area nodes receive energy replenishment while reducing ineffective flight time, the UAV flight distance and speed need to be controlled.

In summary, the overall objective of D-JERDG is to minimize the UAV flight energy consumption and maximize its throughput while minimizing the node death rate. This problem is a multiobjective optimization problem, and the objective functions can be formulated as follows:(17)$$P1:\min(N_{d},E_{u}^{ave},L_{u}^{total})$$

The constraint condition can be formulated as follows:(18)$$\begin{array}{c} 0\leq \sum_{n=0}^{N}t_{n}\leq T \end{array}$$(19)$$\begin{array}{c} \Delta d_{u,ch_{i}^{tar}}<R_{d} \end{array}$$(20)$$\begin{array}{c} \forall \Delta d_{u,cm_{j}}<R_{c} \end{array}$$(21)$$\begin{array}{c} N_{d}^{i}=\left\{\begin{array}{c} 0,RE_{i}\left(t\right)=0\\ 1,RE_{i}\left(t\right)>0 \end{array}\right. \end{array}$$(22)$$\begin{array}{c} v_{\min}\leq v_{u}\left(t\right)\leq v_{\max} \end{array}$$(23)$$\begin{array}{c} 0<\theta_{u}\left(t\right)\leq 2\pi \end{array}$$(24)$$\begin{array}{c} 0\leq x_{u}\left(t\right)\leq L \end{array}$$(25)$$\begin{array}{c} 0\leq y_{u}\left(t\right)\leq L \end{array}$$

In objective functions, $N_{d}^{i}\left(i=\left(1,2,\ldots ,k\right)\right)$ is denoted as the number of dead nodes, $L_{u}^{total}$ represents the total flight distance, and $E_{u}^{ave}$ represents the UAV’s average flight energy consumption. In the constraint condition, Formula (18) represents the UAV with a maximum flight time of no more than *T*. Formula (19) represents the maximum data transmission range of the node into the UAV to establish a communication link with the UAV. Formula (20) indicates that nodes can harvest energy from the UAV within the maximum charging range of the UAV. Formula (21) represents the energy state. If $RE_{i}\left(t\right)=0$, node *i* is considered a dead node. If $RE_{i}\left(t\right)<0$, node *i* is in a normal state. Formulas (22) and (23) represent the size limit of the UAV’s speed and heading angle. Formulas (24) and (25) represent that the UAV’s flight range does not extend beyond the two-dimensional area.

## 4. A Joint Energy Replenishment and Data Collection Algorithm Based on Deep Reinforcement Learning

In this section, we presented the effective implementation of D-JERDG in UAV-WRSN. First, we introduced a UAV-WRSN with a random topology structure. Then, we described our proposed cluster head selection algorithm. Using the obtained set of CH nodes, we applied the simulated annealing algorithm to solve the traveling salesman problem (TSP). Afterward, we provided the state space, action space, and reward function of the MODDPG algorithm.

### 4.1. Cluster Head Selection Algorithm

Referring to [39], this network studied in this paper is a UAV-WRSN that uses a dynamic routing protocol. Figure 2 illustrates the generated clusters and data flow. Compared with static routing protocols, dynamic routing protocols can evenly consume the energy of nodes. However, in a traditional LEACH routing protocol [45], the CM nodes determine whether to become CH nodes based on randomly generated thresholds. For each CM node $cm_{j}$, $w_{j}^{th}$ is given as follows:(26)$$w_{j}^{th}=\left\{\begin{array}{c} \frac{p}{1-p(r\;\mathrm{mod}\;\frac{1}{p})},\mathrm{if}\;j\in cm_{j}\\ 0,\;\;\;\mathrm{otherwise} \end{array}\right.$$
where *p* is the expected percentage of CH nodes, and *r* is the current number of rounds. This method lacks consideration for the remaining energy and location of the nodes. We have redesigned the CH node selection mechanism based on these two factors. Due to insufficient energy information before the network operation, we utilize the K-means algorithm to partition all the data into *m* clusters and identify *m* centroids within each cluster. The overall optimization objective of the K-means algorithm is to minimize the sum of distances between each point and its respective centroid within the clusters. Based on the distances between nodes and centroids, we have determined *m* CH nodes. As a result, when the UAV reaches a CH node, it can cover more nodes, thereby improving the charging efficiency.

At each round, the cluster member nodes at each cluster elect CH nodes based on their own energy status and location information to balance the energy consumption of the nodes. Dynamic routing protocols are used within each cluster. We stipulate no communication link between the sensor nodes and BS. CM nodes use CH nodes as the next hop for communication, while CH nodes collect data sent by member nodes and transmit them to the UAV. (If a CH node has nonzero data storage after the UAV completes a flight cycle, it will continue to transmit data as a CM node in the next mission.) The weight of CM nodes can be obtained using the following formula:(27)$$w_{cm_{j}}=\kappa \frac{E_{\max}-RE_{j}^{cm}\left(t\right)}{RE_{ave}\left(t\right)}+\left(1-\kappa \right)\frac{d_{i,j}}{d_{ave}}$$
where $RE_{ave}\left(t\right)$ represents the average remaining energy of CM nodes, $d_{ave}$ represents the average distance between CM nodes and CH nodes, $d_{i,j}$ represents the distance between the CH node $ch_{i}$ and the CM node $cm_{j}$, and κ is a weighting coefficient. The weight value $w_{j}$ represents the priority or importance of CM; among CM nodes, the one with the highest $w_{j}$ will be selected as the CH node in the next mission round. The CH node will then join the cluster as a CM node. Based on the obtained coordinates of the CH nodes, this paper utilizes the SA algorithm to solve the TSP. The pseudocode for the SA algorithm is presented in Algorithm 1. The specific algorithmic process is shown in Figure 3.

### 4.2. UAV Control Algorithm Based on MODDPG

In contrast with reinforcement learning algorithms based on an action value, our scenario involves a continuous action space for controlling the instantaneous speed and heading angle of the UAV. This necessitates a distinct modeling and solution approach. To tackle this continuous action space, we propose utilizing the MODDPG model for policy learning. The algorithmic specifics of MODDPG are elaborated in Algorithm 2.
**Algorithm 1** Solve TSP by using simulated annealing algorithm**Input:** **The coordinates of CH nodes $l_{i}$****Output:** **The sequence of CH nodes $l_{new}$**  1:Initialize maximum iteration temperature $K_{0}=50,000$, the sequence of CH nodes $l_{i}$, and annealing coefficient q=0.98.  2:$K\leftarrow K_{0}$.  3:**while** $K>K_{0}$ **do**  4:   Swap the order of any two nodes to generate a new visiting sequence $l_{new}$.  5:   Calculating $l_{new}$, denoted as df.  6:   **if** df>0 **then**  7:     **if** $exp\left(\frac{df}{K_{0}}\right)>rand\left(0,1\right)$ **then**  8:        $l_{f}\leftarrow l_{new}$  9:     **end if**10:   **else**11:     K=K×q12:   **end if**13:**end while**

**Algorithm 2** MODDPG
**Input:** 
**State space $s_{rl}$**
**Output:** 
**The action of UAV $a_{rl}$**
  1:Initialize Actor network parameters $\theta^{\mu }$, and Target Actor network parameters $\theta^{\mu^{\prime }}$, $\theta^{\mu^{\prime }}\leftarrow \theta^{\mu }$.  2:Initialize Critic network parameters $\theta^{Q}$, and Target Critic network parameters $\theta^{Q^{\prime }}$, $\theta^{Q^{\prime }}\leftarrow \theta^{Q}$.  3:Initialize experience Replay Buffer.  4:**for** episode=0to *M* **do**  5:   Initialize $s_{rl}$.  6:   **for** time step = $t_{1},t_{2},\ldots ,t_{n}$ **do**  7:     **repeat**  8:        With probability of ϵ choose an action $a_{rl}=clip\left(\mu \left(s_{rl}|Q_{\mu }\right)+\epsilon ,a_{low},a_{high}\right)$.  9:        Perform action $a_{rl}$ and observe reward $r_{rl}$ and next state $s_{rl}^{\prime }$.10:        Store the transition $\left(s_{rl},a_{rl},r_{rl},s_{rl}^{\prime }\right)$ from *P*.11:        **if** *P* ≥ Batch size **then**12:           Randomly sample Mini batch transitions $\left(s_{rl},a_{rl},r_{rl},s_{rl}^{\prime }\right)$ from *P*.13:           Compute $y_{rl}$ (19).14:           Update Critic network by minimizing the critic loss (20).15:           Update Actor network by maximizing the actor loss (21).16:           Soft Update Target Network Parameters.17:           $\theta^{Q^{\prime }}\leftarrow \tau \theta^{Q}+\left(1-\tau \right)\theta^{Q^{\prime }}$           $\theta^{\mu^{\prime }}\leftarrow \tau \theta^{\mu }+\left(1-\tau \right)\theta^{\mu^{\prime }}$18:        **end if**19:     **until** $\sum_{n=0}^{N}t_{n}\geq T$20:   **end for**21:
**end for**



State Space: The state space set is defined as follows:
(28)$$s_{rl}\left(t\right)=\left\{d_{u,ch_{i}^{tar}}^{x}\left(t\right),d_{u,ch_{i}^{tar}}^{y}\left(t\right),x_{u}\left(t\right),y_{u}\left(t\right),N_{f}\left(t\right),N_{d}\left(t\right),\forall i\in \left[1,m\right]\right\}$$Specifically, with a state space size of 6, $d_{u,ch_{i}^{tar}}^{x}$ and $d_{u,ch_{i}^{tar}}^{y}$ represent the relative distances in the horizontal and vertical coordinates between the UAV and the target CH node *i*, and $N_{f}\left(t\right)$ records the cumulative number of times that the UAV has continuously exceeded the restricted area by time *t*.Action Space: At each *t*, the action of the UAV is defined as follows:$\theta_{u}\left(t\right)\in \left(0,2\pi \right]$: Instantaneous heading angle of the UAV along the horizontal direction at *t*.$v_{u}\left(t\right)\in \left[0,v_{max}\right]$: Instantaneous speed of the UAV at *t*.The action of the UAV is defined as $a_{rl}=\left[v_{u}\left(t\right),\theta_{u}\left(t\right)\right]$Reward Function: At any time step $t_{n}$, the UAV receives a reward $r_{rl}$. To ensure the throughput of the UAV, the UAV receives an immediate reward $r_{0}$ whenever it successfully establishes communication with the target node $ch_{i}^{tar}$. The reward function is defined as follows:
(29)$$r_{rl}=\left\{\begin{array}{c} r_{0}\left(t\right)+r_{1}\left(t\right),\;\mathrm{if}\;\Delta d_{u,ch_{i}^{tar}}\left(t\right)\leq R_{d}\\ r_{1}\left(t\right),\;\mathrm{otherwise} \end{array}\right.$$
(30)$$r_{0}\left(t\right)=r_{serve}+\eta_{1}\lambda_{\mathrm{cov}er}\left(t\right)+\eta_{2}R_{ch_{i}^{tar}}^{tran}\left(t\right)$$
(31)$$r_{1}\left(t\right)=-\eta_{2}\Delta d_{u,ch_{i}^{tar}}\left(t\right)-\eta_{3}\left(E_{u}^{pro}\left(t\right)+N_{d}\left(t\right)\right)$$Specifically, the reward function is divided into two parts, $r_{0}$ and $r_{1}$, where $r_{serve}$ represents the reward for the UAV establishing a connection with CH nodes; $\lambda_{cover}$ indicates the number of nodes covered by the UAV; and $\eta_{1}$, $\eta_{2}$, and $\eta_{3}$ are reward weight factors for the optimization objectives. $r_{1}$ is based on the UAV actions at each time step and includes factors such as the distance between the UAV and CH nodes, the energy consumption of the UAV flight, and the number of dead nodes $N_{d}\left(t\right)$.

The MODDPG network framework is shown in Figure 4; it consists of four networks, with the main network and target network each comprising two networks: the actor network, critic network, target actor network, and target critic network. The main network and target network share the same network structure, where the actor network outputs UAV actions, and the critic network evaluates these actions to update the actor network. The MODDPG algorithm optimizes the UAV instantaneous speed and heading angles, ultimately learning the optimal policy.

The calculation of the target value $y_{rl}$ is as follows:(32)$$y_{rl}=r_{rl}+\gamma Q^{\prime }\left({s_{rl}}^{\prime },\mu^{\prime }\left({s_{rl}}^{\prime };\theta^{\mu^{\prime }}\right);\theta^{Q^{\prime }}\right)$$
The critic network is trained using gradient descent to minimize the loss between the predicted value $Q(s_{rl},a_{rl};\theta^{Q})$ and the target value $y_{rl}$.
(33)$$Loss=\frac{1}{M}\sum {\left(y_{rl}-Q\left(s_{rl},a_{rl};\theta^{Q}\right)\right)}^{2}$$

Update the actor network using a gradient-based policy algorithm:(34)$$\nabla_{\theta^{\mu }}J=\frac{1}{M}\sum \nabla_{a_{rl}}Q\left(s_{rl},a_{rl};\theta^{Q}\right){|}_{s=s_{rl},a=\mu \left(s_{rl}\right)}\nabla_{\theta^{\mu }}\mu \left(s_{rl};\theta^{\mu }\right){|}_{s_{rl}}$$
where *J* represents the total discounted cumulative reward. The specific algorithm flow is described as follows: Initialize the replay buffer, critic network, and actor network parameters $\theta^{Q}$ and $\theta^{\mu }$, as well as the target critic network and target actor network parameters $\theta^{Q^{\prime }}$ and $\theta^{\mu^{\prime }}$ (lines 1–3). Perform the exploration phase for the UAV in lines 4–10, and generate an action $a_{rl}=clip\left(\mu \left(s_{rl}|Q_{\mu }\right)+\epsilon \right)$ from the actor network. This process is repeated until the maximum task time is reached, and the experiences are stored in the replay buffer. Updating network parameters, when the experience replay buffer *P* is full, start training the networks to update actor network and critic network parameters (lines 11–15). Update the critic network parameters by minimizing the loss function (line 14), and update the actor network parameters using the deterministic policy gradient (line 15). Finally, use the soft update technique to update the parameters of the target actor network, controlling the update frequency to converge faster toward the optimal policy (lines 17–21).

## 5. Experimental Results

In this section, we conducted simulation results to validate the effectiveness of D-JERDG. Specifically, we provided numerical results and analyzed the convergence of D-JERDG by adjusting the number of CM nodes, CH nodes, and maximum charging radius. We compared D-JERDG with MODDPG and the random method. We analyzed the differences between the algorithms from five aspects: node death rate, data overflow rate, UAV throughput, average flight energy consumption, and time utilization. By comparing these metrics, we were able to evaluate and highlight the differences and advantages of D-JERDG over the other algorithms in terms of efficiency and performance.

D-JERDG: D-JERDG uses the K-means clustering algorithm to divide nodes into multiple clusters, designing the CH node selection mechanism based on an improved dynamic routing protocol. The SA algorithm determines the CH node access sequence, and the UAV flight actions are controlled by inputting the UAV and network states into the DRL model MODDPG.MODDPG (multiobjective deep deterministic policy gradient) [20]: This method selects the node with the highest data volume as the target node. By observing the relative positions of the UAV and the target node, as well as the node states, a DRL model is established to control the UAV learning of all node positions. Through accessing the nodes, data collection and range-based charging are achieved.Random: The method is based on the MODDPG [20] definition, with the difference being that the random algorithm randomly selects a node as the target node and uses the DRL model to optimize the UAV’s flight trajectory.

### 5.1. Simulation Settings

We consider the UAV taking off from the BS, the maximum flight time *T* to be set at 10 min. The sensor nodes are randomly distributed within a 400×400
$m^{2}$ square two-dimensional area. The UAV flies at an altitude of 10 m, with a maximum data transmission radius $R_{d}=10$ m and a maximum charging radius $R_{c}=30$ m. The maximum flight speed of the UAV is set to 20 m/s [20]. The coordinates of the nodes are randomly generated and remain fixed. The nodes’ maximum data storage capacity is $Q_{max}=100$ Mb [40]. The total energy of the nodes is set to $E_{max}=800$ J [40], and their initial energy is randomly generated within the range of [0 J, 800 J] [40]. The transmission energy consumption for CM nodes is $e_{t}=0.4$ J [43], and the reception energy consumption for CH nodes is $e_{r}=0.5$ J [43]. Table 3 and Table 4, respectively, show the network parameters of MODDPG and the main environmental parameters, as specified in reference [20]. The operating system environment of the simulation experiment is TensorFlow 2.5.0, TensorLayer 2.2.5, and Python 3.7 on a Linux server with four NVIDIA 3090 GPUs.

### 5.2. Evaluation Metrics

We introduce five key performance metrics to analyze and compare the optimization effects of D-JERDG on UAV-WRSN in terms of system charging efficiency, data collection efficiency, and UAV flight energy consumption.

Node death rate: The node death rate is determined by the proportion of dead nodes, where node *i* is considered dead when $RE_{i}\left(t\right)=0$. A lower node death rate indicates a higher system charging efficiency. This metric determines which charging strategy can maintain the maximum number of surviving nodes in the experiment.Node data overflow rate: The node data overflow rate is determined by the proportion of nodes with data storage exceeding the maximum capacity, represented as $D_{i}\left(t\right)=100$. This metric measures the effectiveness of UAV data collection and the clustering algorithm.UAV throughput: The UAV throughput is the ratio of the number of nodes accessed by the UAV to the total number of nodes within the maximum flight time. It measures the overall efficiency of the UAV-WRSN system.Average flight energy consumption: The average flight energy consumption is defined as the ratio of the total energy consumption during UAV flight to the maximum flight time. It is a key indicator for evaluating the energy consumption of the UAV-WRSN system.Time utilization rate: The time utilization rate represents the ratio of the hover time to the flight time of the UAV during a flight mission. A lower time utilization indicates a higher proportion of time occupied by UAV flights, serving as a metric for evaluating the data collection efficiency of the UAV. The time utilization rate $R_{tu}$ is given as follows:
(35)$$R_{tu}=\frac{t_{hover}}{t_{fly}}\left(T=t_{hover}+t_{fly}\right)$$

### 5.3. Convergence and Stability

After D-JERDG is implemented, we first test the convergence and stability of the proposed model. The episode was set as 400, and the model of the number of different clusters, the number of nodes in each cluster, and the maximum charging radius of the UAV were trained. “C20N200R30” means that there are 20 CH nodes and 180 CM nodes, the maximum charging radius of UAV is $R_{c}=30$ m, and the other sections in this chapter follow the same pattern. Then the functional relationship between episode and reward is shown in Figure 5. As can be seen from Figure 5, the reward of most models increased rapidly before 150 episodes. After 200 episodes, the value of rewards becomes stable and oscillates around a certain value until the training ends. The results indicate that the trained D-JERDG algorithm successfully converges and can make high-reward decisions in a dynamically changing network state. Figure 5a shows the reward curves under different numbers of CH nodes and CM nodes, ranging from “C20N200R30” to “C40N400R30” experimental scenarios. Figure 5b displays the reward curves for different numbers of CH nodes, with the experimental scenarios ranging from “C15N200R30” to “C30N200R30”. Figure 5c illustrates the reward curves under different UAV charging radii, with the experimental scenarios ranging from “C20N200R15” to “C20N200R30”.

Additionally, Figure 6 displays the loss curve and temporal difference error (TD error) curve of network training for the three algorithms, with the experimental scenario being “C20N200R30”. In Figure 6a, the curve trends of the three algorithms are mostly consistent before 150 episodes. After that, due to the increased UAV throughput in D-JERDG, some fluctuations occur. However, overall, the loss curves of the three algorithms converge before 400 episodes. In Figure 6b, we show the TD error curve of network training for the three algorithms. Overall, all three algorithms can converge before 400 iterations. However, before 100 episodes, D-JERDG has a slower convergence rate compared with the other two algorithms. Because we expect D-JERDG to learn the positions of more nodes in the early training episodes, we have implemented a strategy where the set of CH nodes initialized using the K-means algorithm is not fixed. The purpose of this approach is to ensure that D-JERDG can learn the positions of as many nodes as possible. By flexibly adjusting the CH node set, D-JERDG can better adapt to the varying node distributions in different environments, enhancing its performance and training effectiveness. After 100 episodes, as the rewards for D-JERDG start to increase and gradually converge, it also exhibits convergence in the TD error curve.

### 5.4. Performance Comparison

Specifically, without loss of generality, we conducted experiments with different numbers of nodes and varying UAV flight speeds. The number of nodes was set from “C10N100R30” to “C40N400R30”, with a maximum data transmission radius of $R_{d}=10$ m, a maximum charging radius of $R_{c}=30$ m, and a maximum flight speed ranging from 15 m/s to 20 m/s. With an increasing number of nodes, both the energy consumption and data volume of CH nodes increase within the same period. A larger flight speed allows the UAV to reach nodes faster, but it also increases energy consumption. On the other hand, a smaller speed can save energy but may sacrifice some throughput and result in a certain number of dead nodes, thereby affecting UAV throughput, the number of dead nodes, and time utilization. Additionally, as the number of nodes increases, in our proposed approach, the UAV flight time and distance are influenced by the increasing number of CH nodes, which also affects flight energy consumption. Based on these hypotheses, we conducted comparative experiments.

#### 5.4.1. Node Death Rate

In this section, we compared the performances of different algorithms in terms of node death rate. Specifically, in the MODDPG experiment, we designated the node with the least remaining energy as the target node. The experimental results are shown in Figure 7.

Figure 7a demonstrated a clear disparity in node death rates between D-JERDG and Random. Initially, MODDPG exhibited lower node death rates than D-JERDG for up to 200 nodes. This can be attributed to the advantage of the greedy-based node charging strategy in scenarios with a small number of nodes and low density. However, after reaching 200 nodes, D-JERDG demonstrated a lower node death rate than MODDPG, with a declining trend beyond 300 nodes. When the number of nodes is 250, compared with the MODDPG and random, D-JERDG can decrease the node death rate by about 17% and 44%, respectively. This improvement can be attributed to the clustering algorithm employed in D-JERDG, where several nodes closest to the CH node are selected as CM nodes. This approach ensures that the UAV covers more nodes during each hover, enhancing charging efficiency. The effectiveness of the proposed CH node selection mechanism was also validated, as it increased the likelihood of selecting CM nodes farther from the current CH node as new CH nodes. This enabled energy replenishment and reduced the node death rate of remote area nodes, making D-JERDG more suitable for large-scale networks aiming to maintain network connectivity. From Figure 7b, we examined the variations in node death rates across different algorithms while considering different UAV flight speeds, with 200 nodes in the experiment. Overall, increasing the UAV flight speed resulted in reduced fluctuations in the node death rate, and D-JERDG consistently outperformed MODDPG and random by maintaining lower node death rates. When the speed of the UAV was 15 to 20, compared with MODDPG and random, the average decline rate of D-JEGDG in terms of node death rate was about 10% and 45.5%, respectively. Furthermore, as the flight speed increased, the node death rate exhibited a significant decrease. This highlights the advantageous performance of D-JERDG in terms of node death rate when compared with the other algorithms.

#### 5.4.2. Node Data Overflow Rate

In this section, we analyze the differences in node data overflow rates among different algorithms. In the MODDPG experiment, we designate the node with the highest data volume as the target node, and the experimental results are shown in Figure 8. In Figure 8a,b, with an increasing number of nodes, compared with MODDPG and random, the average decline rate of D-JEGDG in terms of node overflow rate is about 46.4% and 48.2%, respectively. With an increasing speed of the UAV, it is about 35.1% and 46%, respectively. This is because the dynamic routing protocol periodically selects CM nodes as CH nodes, allowing CH nodes to gather data from CM nodes and reduce node data overflow. However, for MODDPG and random, the one-to-one data collection approach leads to lower efficiency and inevitable node data overflow.

#### 5.4.3. UAV Throughput

In this section, we compare the performances of different algorithms in terms of UAV throughput by changing the number of CH nodes and CM nodes in the network. The experimental results are shown in Figure 9. In Figure 9a, we observe that as the number of nodes increases, all algorithms experience a decrease in UAV throughput. However, D-JERDG consistently achieves the highest throughput among the algorithms. Compared with MODDPG and random, the average growth rate of D-JEGDG in terms of UAV throughput is about 79.3% and 43%, respectively. In D-JERDG, the use of a K-means-based clustering algorithm for CH node selection introduces randomness, allowing the UAV to quickly learn the positions of all nodes by obtaining different node coordinates in each training round. This enables D-JERDG to achieve a higher node access count within the same flight time. At 250 nodes, D-JERDG reaches a node access count of 50, indicating that it completes two rounds of CH node traversal. Beyond 300 nodes, D-JERDG throughput reaches a plateau and remains competitive with MODDPG. This can be attributed to the increased number of CH nodes requiring the UAV to spend more time hovering for data collection, leading to longer flight times and a decrease in node access count. In Figure 9b, we analyze the impact of different UAV flight speeds on throughput with a fixed number of 200 nodes. Overall, compared with MODDPG and random, D-JERDG outperforms MODDPG and random algorithms, and the throughput significantly increases as the flight speed rises. The average growth rate of D-JEGDG in terms of UAV throughput is about 55.3% and 42.4%, respectively.

#### 5.4.4. Average Flight Energy Consumption

In this section, we analyze the differences in average UAV flight energy consumption among three algorithms. The experimental results are depicted in Figure 10. In Figure 10a, we observe that the UAV energy consumption in D-JERDG initially increases and then decreases as the number of nodes increases. Before reaching 200 nodes, the larger distances between CH nodes prompt the UAV to adapt its flight speed to minimize the flight time and efficiently collect data, resulting in an increasing energy consumption curve. However, after 200 nodes, as the number and density of CH nodes increase, the distances between them become shorter. Compared with MODDPG and random, the average decline rate of D-JEGDG in terms of average flight energy consumption is about 4.2% and 4%, respectively. When the number of nodes is 350, D-JERDG can decrease the average flight energy consumption by about 10% and 9.8%, respectively. As a result, the UAV adjusts its flight speed by reducing it, which leads to longer flight times but decreases the overall UAV energy consumption. Compared with MODDPG and random, in Figure 10b, we examine the impact of different flight speeds on average energy consumption under the three algorithms. Overall, there is no significant difference among the algorithms (D-JERDG, MODDPG, and random) in terms of average energy consumption. However, as the flight speed increases, the average energy consumption increases as well. Notably, D-JERDG consistently maintains a lower average UAV flight energy consumption compared with MODDPG and random algorithms under different flight speed conditions.

#### 5.4.5. Time Utilization Rate

In this section, we compare the differences in time utilization rates between the D-JERDG and MODDPG algorithms. The experimental results are presented in Figure 11. In Figure 11a, we observe that the time utilization of D-JERDG initially increases and then decreases as the number of CH nodes varies. This trend aligns with the analysis discussed earlier. In general, D-JERDG exhibits a similar pattern to MODDPG, and after 300 nodes, their time utilization becomes comparable. When the number of nodes is 150 to 350, the average growth rate of D-JEGDG in terms of time utilization rates is about 23%. The reason behind this behavior lies in the D-JERDG method, which utilizes the SA algorithm to adjust the sequence of CH node visits, thereby reducing the UAV flight distance and time. Before reaching 300 nodes, with a smaller network size, the total flight time includes longer hover times. At 200 nodes, the time utilization reaches its peak, which aligns with the trend observed in the average UAV energy consumption curve. However, beyond 300 nodes, as the number of CH nodes increases, the flight time also increases. With the total flight time remaining constant, the time utilization decreases accordingly. In Figure 11b, we examine the impact of UAV flight speed on time utilization under both algorithms. Notably, D-JERDG consistently achieves higher time utilization compared with MODDPG, maintaining the highest level throughout the different flight speeds. As the speed of UAV increases, the average growth rate of D-JEGDG in terms of time utilization rates is about 36%.

## 6. Conclusions

This study proposes a DRL-based method called D-JERDG for joint data collection and energy replenishment in UAV-WRSN. The main problems D-JERDG addresses are individual nodes’ low data collection efficiency and the imbalance in node energy consumption. D-JERDG optimizes the network to tackle these issues by considering node inefficiency, UAV flight energy consumption, and UAV throughput. The clustering algorithm based on K-means and an improved dynamic routing protocol is used to cluster the nodes in the network. CH nodes are selected based on the remaining energy and geographical locations of the nodes within the clusters, effectively adapting to the dynamic nature of node energy consumption. Furthermore, a simulated annealing algorithm is employed to determine the visiting sequence, and the DRL model MODDPG is introduced to control the UAV for node servicing. Through extensive simulation results, the evaluation metrics of node inefficiency, UAV throughput, and average flight energy consumption are used to assess the performance of D-JERDG. The results demonstrate that D-JERDG achieves joint optimization of multiple objectives. It outperforms the existing MODDPG approach by significantly reducing node inefficiency, saving flight costs, and improving data collection efficiency. Moreover, multiple research studies suggest that multiagent systems have significant advantages in accomplishing complex tasks. Therefore, Multi-UAV-WRSN is expected to be a primary research focus in the future.

## Figures and Tables

**Figure 1 sensors-24-02386-f001:**
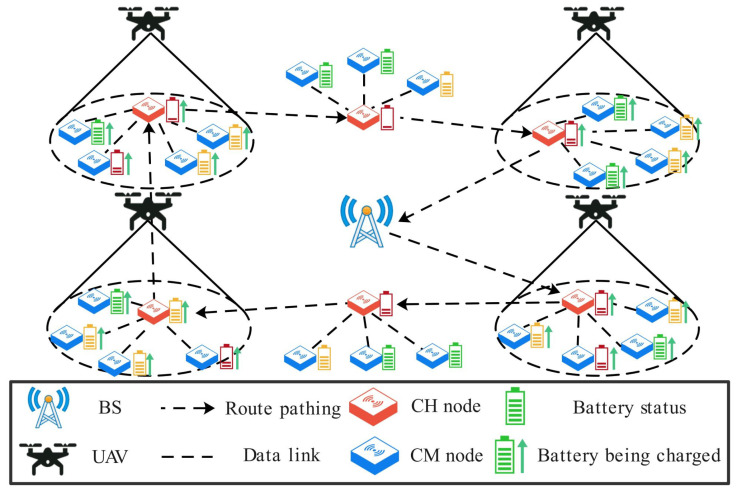
Network model of the UAV-WRSN.

**Figure 2 sensors-24-02386-f002:**
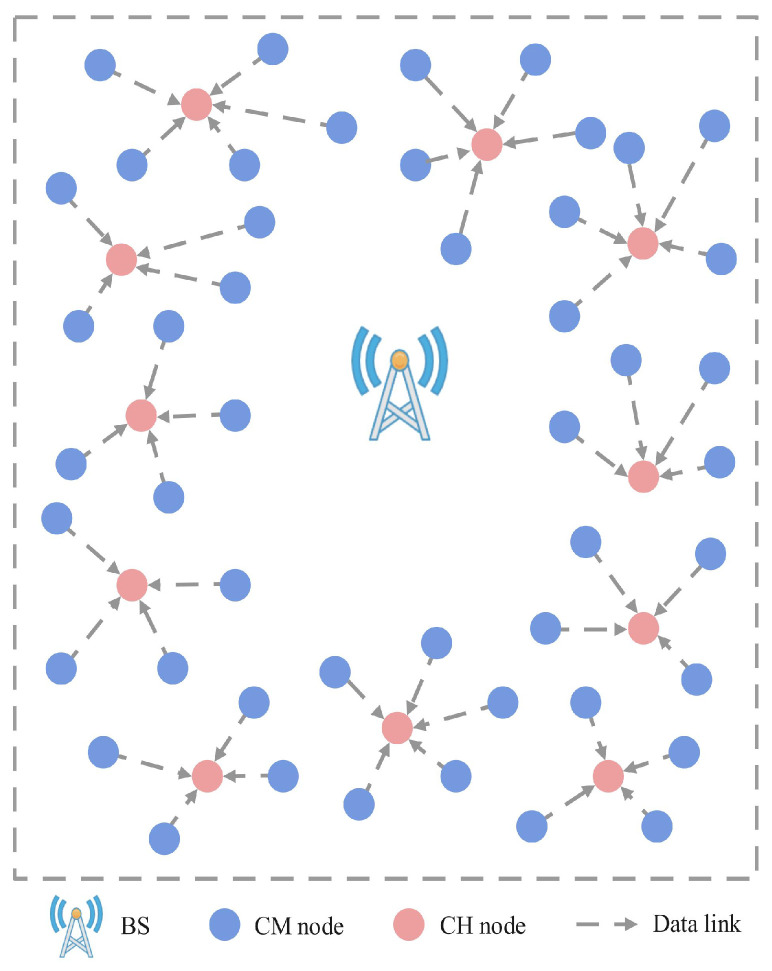
Illustration of clusters and data flow.

**Figure 3 sensors-24-02386-f003:**
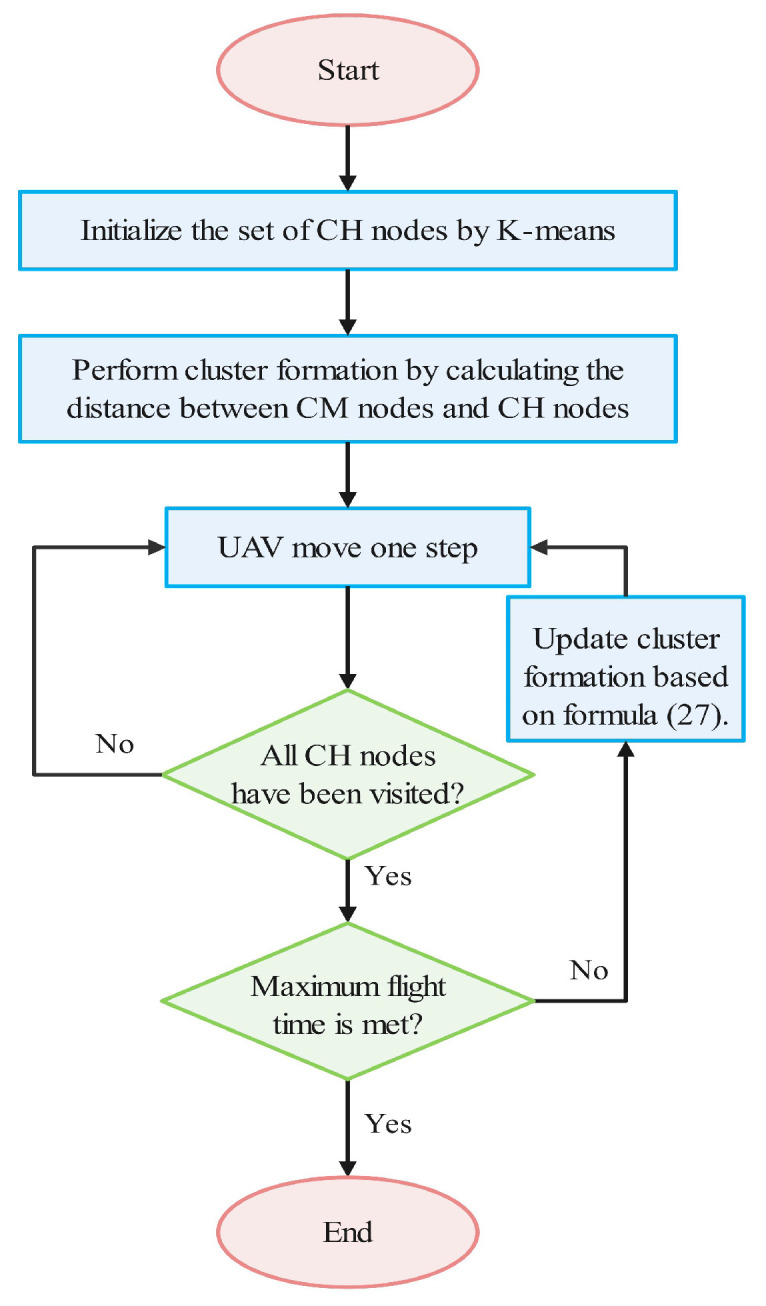
Illustration of the algorithm process.

**Figure 4 sensors-24-02386-f004:**
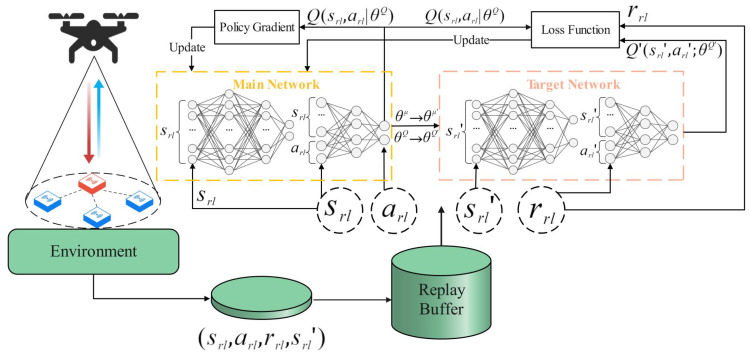
MODDPG network framework.

**Figure 5 sensors-24-02386-f005:**
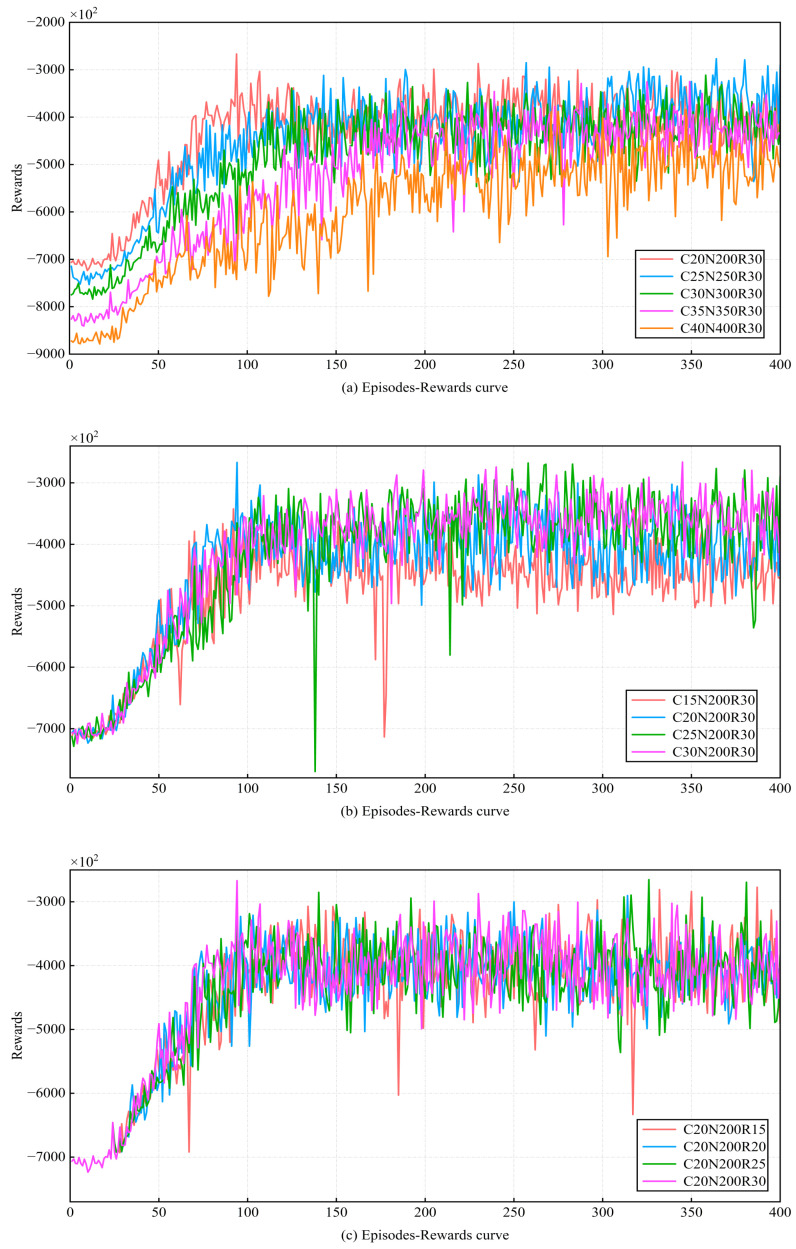
Learning curve with the episode = 400.

**Figure 6 sensors-24-02386-f006:**
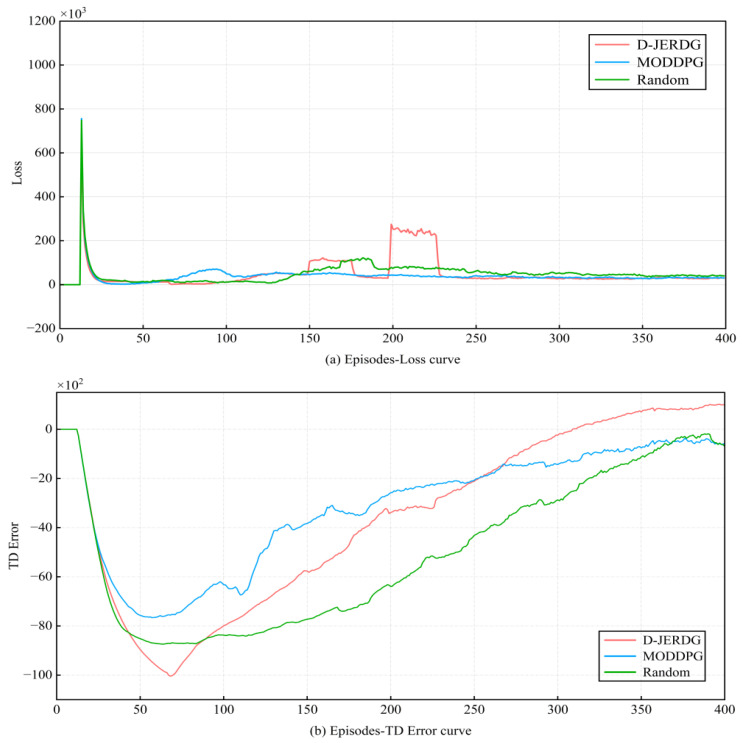
Learning curve with the episode = 400.

**Figure 7 sensors-24-02386-f007:**
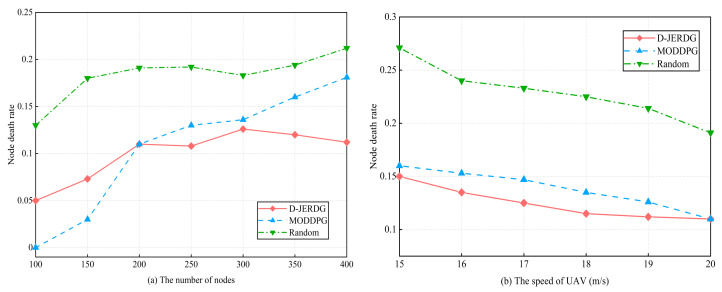
Experimental results of node death rate.

**Figure 8 sensors-24-02386-f008:**
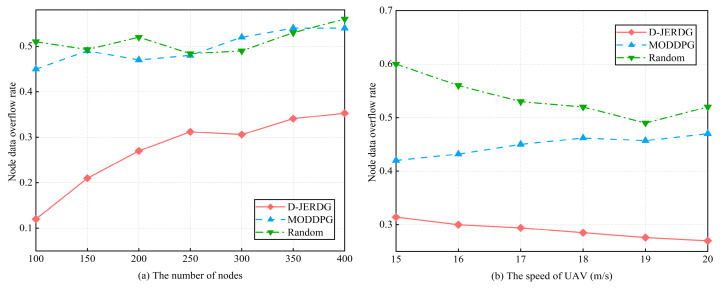
Experimental results of node data overflow rate.

**Figure 9 sensors-24-02386-f009:**
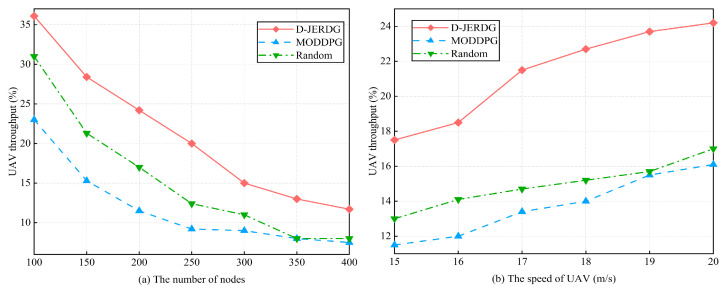
Experimental results of UAV throughput.

**Figure 10 sensors-24-02386-f010:**
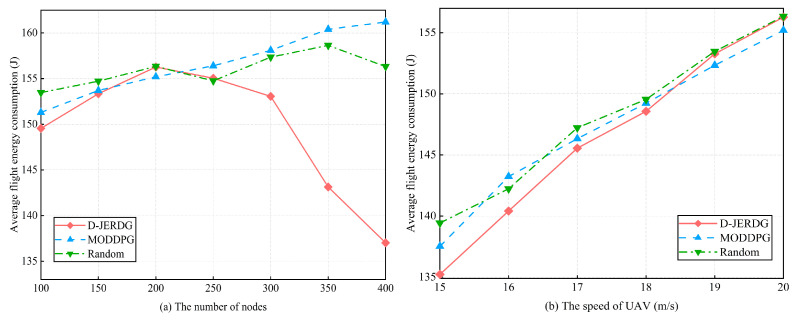
Experimental results of average flight energy consumption.

**Figure 11 sensors-24-02386-f011:**
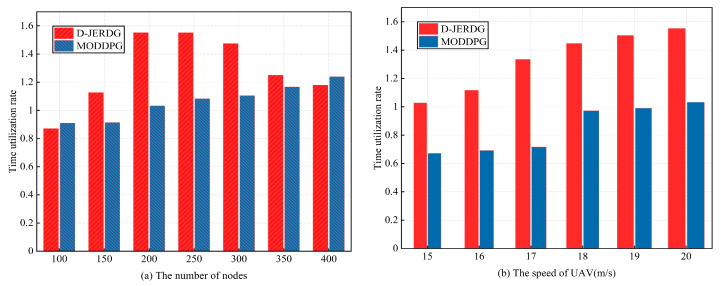
Experimental results of time utilization rate.

**Table 1 sensors-24-02386-t001:** Comparison of related works with D-JERDG.

References	Optimization Objective	Optimization Scheme	Learning-Based	Charging Mode
[20]	Minimize UAV energy consumption while maximizing UAV throughput	DRL	Yes	One-to-multiple
[32]	Minimize hovering points and flying distance of UAV	Particle swarm optimization	No	One-to-multiple
[33]	Maximize energy efficiency	Polynomial-time approximation	No	One-to-one
[34]	Minimize trajectory path while maximizing network lifetime	Ant colony algorithm	No	One-to-one
[35]	Maximize energy of nodes	Geometry-based algorithm	No	One-to-one
[40]	Minimize the overall data packet loss	DRL	Yes	One-to-multiple
[41]	Minimize charging path	DRL Dynamic routing	Yes	One-to-one
[42]	Minimize death time of nodes and UAV energy consumption	DRL	Yes	One-to-one
[43]	Minimize node death rate	DRL	Yes	One-to-one
D-JERDG	Minimize node death rate and flight energy consumption while maximizing UAV throughput	DRL Dynamic routing	Yes	One-to-multiple

**Table 2 sensors-24-02386-t002:** Notation and definition.

Notations	Definitions	Notations	Definitions
*k*	total number of nodes	$D_{i}\left(t\right)$	buffer length of node *i*
*m*	number of CH nodes	$v_{u}\left(t\right)$	speed of the UAV
*h*	number of CM nodes	$\theta_{u}\left(t\right)$	yaw angle of the UAV
$Q_{max}$	buffer capacity of nodes	α	weight coefficient of CM nodes
$E_{max}$	battery capacity of nodes	$R_{c}$	maximum charging distance
$RE_{i}\left(t\right)$	remaining energy of node *i*	$R_{d}$	maximum data transmission distance

**Table 3 sensors-24-02386-t003:** Network parameters.

Parameters	Values
Episodes (*M*)	400
Actor network structure	400×300×300
Critic network structure	400×300
Actor network hidden layers	3
Critic network hidden layers	2
Batch size	64
Learning rate for actor $\left(lr_{a}\right)$	$1\times 10^{-3}$
Learning rate for critic $\left(lr_{c}\right)$	$1\times 10^{-3}$
Discount factor (γ)	0.9
Replay buffer (P)	8000
Target network soft update rate (τ)	$1\times 10^{-3}$
Reward weights $\left(\eta_{1},\eta_{2},\eta_{3}\right)$	50,100,5

**Table 4 sensors-24-02386-t004:** Main simulation parameters.

Parameters	Values
Node transmit power $\left(P_{t}\right)$	$1\times 10^{-3}$ W
Buffer capacity of nodes $\left(Q_{max}\right)$	100 Mb
Battery capacity of nodes $\left(E_{max}\right)$	800 J
The initial energy of nodes	rand(0 J, 800 J)
The initial buffer length of nodes	rand(0 Mb, 5 Mb)
Weighting coefficient (κ)	0.7
Channel bandwidth (B)	1 MHz
Channel power gain ($\gamma_{0}$)	−30 dB
Noise power $\left(\sigma^{2}\right)$	−90 dBm
NLoS path loss coefficient $\left(\eta^{NLos}\right)$	0.2
Los probability coefficient (α,β)	10,0.6
Blade profile power $\left(P_{0}\right)$	79.8563 W
Induced power $\left(P_{f}\right)$	88.6279 W
Tip speed of rotor blade $\left(U_{tip}\right)$	120 m/s
Mean rotor induced velocity $\left(v_{0}\right)$	4.03 m/s
Fuselage drag ratio $\left(d_{0}\right)$	0.6
Air density $\left(\rho_{air}\right)$	1.225 km/$m^{3}$
Rotor solidity $\left(S_{rotor}\right)$	0.05
Rotor disc area $\left(A_{rotor}\right)$	0.503 $m^{2}$

## Data Availability

The data are not publicly available due to it is not permitted.

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
