# Peer review of "Deep-Reinforcement-Learning-Based Joint Energy Replenishment and Data Collection Scheme for WRSN"

_sensors, 2024, doi:10.3390/s24082386_

Round 1

Reviewer 1 Report

Comments and Suggestions for Authors

The authors introduced and tested a new approach called D-JERDG for Wireless Rechargeable Sensor Networks (WRSN), which shows superior performance in preliminary experiments compared to existing the MODDPG and the Random methods.

Below my comments:

1) The authors forgot from specifying the method they are referring to in the statement: “Compared to traditional charging approaches, this method fundamentally overcomes the predicament of nodes relying solely on battery power.”

2) Authors must first provide the full terms for BS, SN, MODDPG, and D-JERDG, followed by their abbreviations in parentheses, and use the abbreviations consistently thereafter.

3) The authors need to clarify why they state that nodes only consume energy during data reception and transmission, since even in sleep mode they still consume minimal energy.

4) References to Shannon's Formula, Simulated Annealing and Line 207 must be included by the authors.

5) Authors must explain their reasons for choosing the K-Means algorithm for node clustering.

6) There is a need for clarification on how the UAV (Unmanned Aerial Vehicle) obtains the positions of all nodes.

7) Authors should provide more complete details about the neural networks used in their study, including the names of the networks, examples of input data for the two neural networks, and the training results of these networks.

8) In line 23, the authors compared D-JERDG to traditional charging approaches, without providing any indication that MODDPG and the Random method are considered traditional charging approaches.

Reviewer 2 Report

Comments and Suggestions for Authors

The authors of this manuscript propose a reinforcement learning based approach for UAV-WRSN to optimize the system overall performance.

Some minor concerns are as follows, mainly regarding experimental results analysis:

1. In the result analysis, two algorithms other than the proposed were used for performance comparison: MODDPG algorithm and Random. Is "Random" algorithm a proper noun? If yes please provide the reference, otherwise explain in general what type of "random" factors you introduced.

2. Notation in Section 5.1. I assume when you talk about energy in Section 5.1, the unit is "Joules", so please make them capital. Otherwise it could: (1) mess up with notation "j" in your equations, (2) not being correctly used as unit for energy.

3. In Figure 8, are you using normalized UAV throughput? If yes please mention that in Section 5.4.3 and talk about how you normalize it. Otherwise please have your Y-axis of Figure 8 labeled with unit.

4. Combine Figure 9 and Section 5.4.4, I assume you are talking about power consumption instead of energy consumption? If yes please consider to use the term "power consumption", otherwise the unit shown on Y-axis of Figure 9 should be for energy (Joules) rather than power (Watt).

5. For Section 5.4.5, please briefly explain how you calculate the time utilization rate.

Comments on the Quality of English Language

Writing is fine, no significant fluency issue identified.

Some typos and proofreading needed. See below for an incomplete list.

1. The abbreviation "MODDPG" was not shown in full when first introduced (line12, page 1, Abstract)

2. Same for D-JERDG (line4, page 1, Abstract).

3. "[0j − 800j]" should be "[0j, 800j]" (line363, page 11, Section 5.1)

Reviewer 3 Report

Comments and Suggestions for Authors

In this paper, the authors propose D-JERDG, a joint energy replenishment and data collection method for WRSNs based on deep reinforcement learning (DRL). Extensive simulation results show that the proposed D-JERDG achieves joint optimization of multiple objectives and exhibits significant advantages over the baseline in terms of throughput, time utilization, and charging cost, among other indicators.

The paper is well written and readable form. I advise the authors to carryout revision as per the below comments:

1. Include some statistical figures to highlight the % improvement of the proposed scheme when compared to other existing schemes.

2. Include a table for related works.

3. 3.1.2. Sensor Node Model shall be listed as points one by one.

4. In transmission model, I found interference model is missing.

5. Section 2, eq 12b and 12c, why it is equality rather than inequality?

6. In table 3, where is the maximum battery capacity or initial energy parameters?

7. Explain the trade-offs with node death rate and UAV speed.

8. Where is the optimization done? I mean the authors should justify for what value of the number of nodes, the death rate is minimum and speed is also high with minimized energy? If I am wrong, kindly justify your claim.

Comments on the Quality of English Language

Minor editing of English language required

Round 2

Reviewer 3 Report

Comments and Suggestions for Authors

The authors have carried out all suitable corrections suggested by the reviewers and they have improved the paper well. Hence the paper shall be accepted in present form.